# Neuronal Repressor REST Controls Ewing Sarcoma Growth and Metastasis by Affecting Vascular Pericyte Coverage and Vessel Perfusion

**DOI:** 10.3390/cancers12061405

**Published:** 2020-05-29

**Authors:** Zhichao Zhou, Yuanzheng Yang, Fei Wang, Eugenie S. Kleinerman

**Affiliations:** Department of Pediatrics, The University of Texas, MD Anderson Cancer Center, Houston, TX 77030, USA; zczhou@hotmail.com (Z.Z.); yyang9@mdanderson.org (Y.Y.); FWang8@mdanderson.org (F.W.)

**Keywords:** REST, tumor vasculature, pericytes, Ewing sarcoma

## Abstract

Survival rates for Ewing sarcoma (ES) patients with metastatic disease have not improved in over 20 years. Tumor growth and metastasis are dependent on tumor vasculature expansion; therefore, identifying the regulators that control this process in ES may provide new therapeutic opportunities. ES expresses high levels of repressor element 1 silencing transcription factor (REST), which is regulated by the EWS-FLI-1 fusion gene. However, the role of REST in ES growth and the regulation of the tumor vasculature have not been elucidated. To study this role, we established REST-knockout human TC71 ES cell lines through CRISPR/Cas9 recombination. While knockout of REST did not alter tumor cell proliferation in vitro, REST knockout reduced tumor growth and metastasis to the lung in vivo and altered tumor vascular morphology and function. Tumor vessels in the REST-knockout tumors had a punctate appearance with significantly decreased tumor vascular pericytes, decreased perfusion, and increased permeability. REST-knockout tumors also showed increased apoptosis and hypoxia. These results indicate that REST plays a critical role in ES vascular function, which in turn impacts the ability of ES tumors to grow and metastasize. These findings therefore provide a basis for the targeting of REST as a novel therapeutic approach in ES.

## 1. Introduction

Ewing sarcoma (ES) is the second most common malignant bone tumor in children and young adults, and the lung is the most common site of metastasis. Increasing the dose and frequency of chemotherapy administration has improved the survival rate to 75% for patients with a tumor in an extremity and no detectable metastases [1,2,3,4,5]. However, there have been no major advancements in treatment for more than 10 years. The survival for patients presenting with metastases is much worse (<25%) and has not improved in over 30 years [6,7]. Salvage chemotherapy protocols for relapsed disease are ineffective, as patients typically die within 1 year of relapse. Tumor vessels play an important role in providing nutrients and oxygen to the tumor, which are required for sustained growth. Understanding the mechanisms that contribute to the successful formation of functional tumor vessels in ES and understanding how ES tumor cells contribute to this process, may allow the identification of new therapeutic targets that inhibit vascular expansion.

The origin of ES is still controversial. Several studies suggest that its origin is in the neural crest [8,9,10]. This origin would indicate that genes that play a role in neuronal tumors may also contribute to the tumorigenesis of ES. Repressor element 1 silencing transcription factor (REST) is a neuronal repressor gene that regulates neuronal stem cell differentiation [11,12]. REST also plays a multifunctional role in the regulation of non-neurogenic cells [13,14]. In tumor growth, REST has a dual function depending on the cellular context. We have previously shown that REST is upregulated in both ES cell lines and patient samples and that EWS-FLI-1, the hallmark fusion gene protein in ES tumors, regulates REST expression [15]. We also showed that inhibition of REST expression by shRNA reduced tumor growth and increased tumor hypoxia and apoptosis without modifying the expression of EWS-FLI-1 or decreasing cell proliferation in vitro. Inhibiting REST also altered the morphology of the tumor vessels. While neither CD31 nor VEGF expression in the REST shRNA ES tumors was decreased, tumor vascular pericyte coverage decreased significantly. This decrease was associated with increased hypoxia and tumor cell apoptosis. Pericytes are important for vascular stabilization and efficient blood flow, suggesting that REST in ES is critical for maintaining vascular perfusion and that the decreased tumor growth in the REST shRNA tumors was due to an effect on the tumor vasculature rather than a direct effect of the tumor cells.

The REST shRNA expression vector that we used previously decreased REST expression by 50–70% [15]. However, we were unable to completely downregulate the gene or maintain complete inhibition in vivo. To confirm and extend these investigations focusing on the role of REST in tumor vascular expansion and function, we used CRISPR/Cas9 technology to knock out REST expression in ES cells. In addition to analyzing the effect of REST knockout (KO) on tumor growth and vascular morphology, we evaluated the effect of REST KO on lung metastasis, vascular perfusion, and vascular permeability. These data confirmed that REST plays a critical role in maintaining and expanding ES tumor vessels that are required for tumor growth. These findings together with our previous studies show that interfering with vascular formation and expansion severely retards not only tumor growth but also metastasis to the lung [16,17,18,19,20,21]. REST is therefore a potential target for ES therapy.

## 2. Results

### 2.1. Establishment of CRISPR/Cas9–REST-KO Clones

We previously showed that inhibiting REST expression in TC71 and A4573 ES cells using shRNA slowed tumor growth and altered vascular morphology [15]. To further investigate the effects of REST on ES tumor vasculature, TC71–REST-KO clones were generated using CRISPR/Cas9 recombination. Five single-guide RNA (sgRNA) oligonucleotides targeting five specific human REST genomic regions were synthesized and cloned into CRISPR/Cas9 vector pX458-U6-chimeric_BB(+85)-CBh-NLS-hSpCsn1-2A-GFP. The sgRNA expression clones were verified by restriction enzyme digestion and sequencing. TC71 cells were either transfected with a single sgRNA expression vector for single-nicking recombination or cotransfected with two sgRNA expression vectors for double-nicking recombination. Double-nicking CRISPR/Cas9 by cotransfection of two sgRNA expression vectors had much higher knockout efficiency than single-nicking recombination (Appendix A). Western blot analysis confirmed that REST protein expression was dramatically reduced in several different CRISPR/Cas9–REST-KO clones compared with DAOY positive control cells and the TC71 parental cells; densitometry analysis demonstrated that REST protein expression was decreased by at least 80% in the R1106 and R1606 REST-KO cells compared with the parental cells (Figure 1A).

Immunofluorescence staining confirmed that REST expression was significantly inhibited in the R11106 and R1606 cells compared with the TC71 parental and TC71 CRISPR/Cas9 recombination clone with normal expression of REST (RC-control) cells (Figure 1B).

### 2.2. Effect of Down-Regulation of REST on Cell Proliferation In Vitro and Tumor Growth In Vivo

The inhibition of REST had no effect on cell proliferation. The in vitro doubling time did not significantly differ between TC71 parental cells, RC-control cells, and REST-KO clones R1106 and R1606: 22.0 h (standard deviation, ±5.0), 26.0 ± 5.0 h, 21.7 ± 6.9 h, and 26.7 ± 6.1 h, respectively.

To determine the effect of REST KO in vivo, TC71 parental cells, RC-control cells, and R1106 or R1606 REST-KO cells were injected into the tibias of mice. Twenty-four days after injection, R1106 and R1606 REST-KO tumors were significantly smaller on average than RC-control or TC71 parental tumors (Figure 2A). Immunofluorescence staining of the tumor samples for REST expression confirmed that REST was significantly reduced in the R1106 and R1606 tumors compared with the RC-control tumors (Figure 2B). Expression of the proliferation marker Ki67 was also quantified in the tumor samples. Echoing our in vitro findings, the Ki67 expression levels did not significantly differ between the REST-KO tumors and the RC-control tumors (Figure 2C). These results suggest that the decreased growth in R1106 and R1606 REST-KO tumors was not the result of decreased tumor cell proliferation. R1106 and R1606 REST-KO tumors also showed decreased metastatic potential to the lung (Figure 2D).

### 2.3. Effect of REST KO on Tumor Vascular Morphology

Tumor growth requires a robust functional vasculature. Since the growth of ES tumors in vivo was not explained by an effect on tumor cell proliferation, we next evaluated the tumor vasculature morphology in the R1106 and R1606 REST-KO and RC-control tumors, first by staining with CD31. Vessels in the RC-control tumors showed open lumens, whereas the vessels in the R1106 and R1606 REST-KO tumors had a punctate morphology (Figure 3A, white arrows).

Since pericytes are critical for vascular stabilization and maintaining an open lumen, we next evaluated tumor vessel pericyte coverage. Using CD31 to identify tumor vessels and the pericyte marker NG2, we evaluated both the total number of pericytes in tumors and the ratio of pericytes to tumor vessel cells (Figure 3B, yellow color highlighted by white arrows identifies the double positive staining). The total number of pericytes was significantly reduced in the REST-KO tumors compared with the RC-control tumors (Figure 3B,C). In addition, the ratio of NG2-positive to CD31-positive cells was significantly decreased in the REST-KO tumors (Figure 3B,D). These findings suggest that REST regulated the tumor vascular morphology by decreasing pericyte coverage.

### 2.4. Down-Regulation of REST Reduces Tumor Vascular Perfusion and Increases Permeability

Decreased pericyte coverage on tumor vessels can affect vascular perfusion and permeability. Since REST KO was associated with a decreased NG2:CD31 ratio (Figure 3D), we next evaluated vascular function by assessing the effect of REST KO on perfusion and vascular permeability in the R1606 tumors and RC-control tumors. A DyLight 594–labeled lectin perfusion assay was used to evaluate the effect of REST down-regulation on tumor vascular perfusion. Consistent with our data in Figure 3A, vascular morphology was altered in the R1606 REST-KO tumors; the tumor vessels had a punctate appearance with few open lumens (Figure 4A). In addition, the mean perfused area in the R1606 REST-KO tumors was significantly decreased compared with that in RC-control tumors (Figure 4B, *p* < 0.01). To determine the percentages of functional perfused and non-functional vessels in the different tumor samples, CD31 staining (fluorescein isothiocyanate (FITC)) was used to identify all vessels, and co-localization of lectin and CD31 was used to identify functional vessels (Figure 4C, yellow color indicated by white arrows shows the double positive staining). The percentage of functional perfused vessels among all vessels in R1606 REST-KO tumors was significantly decreased compared with that in RC-control tumors (Figure 4D, *p* < 0.01).

These data demonstrate that down-regulation of REST decreased tumor vascular perfusion.

In addition to decreasing vascular perfusion, decreased pericyte coverage increases vascular permeability, which is an indication of decreased vascular function. To determine whether REST KO affected vascular permeability, four weeks after tumor cell injection mice bearing TC71-RC-control, R1106, and R1606 tumors were injected intravenously with green FITC-labeled dextran (which leaks from permeable vessels into the tumor tissues) five minutes before the mice were sacrificed. Normal vessels are not permeable to this high molecular weight dextran. Decreased vascular pericytes will result in increased vascular permeability, which in turn will increase the leakage of the FITC labeled dextran into the tumor tissues. CD31 staining of the tumor samples also was performed as before. CD31-positive vessels in R1106 and R1606 REST-KO tumors again had a punctate and irregular morphology compared with the RC-control tumors (Figure 5A). FITC-labeled dextran was increased in R1106 and R1606 tumors compared with RC-control tumors. Quantitative analysis confirmed that down-regulation of REST increased tumor vessel permeability, as FITC-labeled dextran was significantly increased in R1106 and R1606 tumors compared with RC-control tumors (Figure 5B, *p* < 0.01).

### 2.5. REST KO Increased Tumor Hypoxia and Apoptosis

Decreased vascular perfusion and increased vessel permeability have been shown to increase tumor tissue hypoxia and apoptosis. We therefore quantified tumor hypoxia and apoptosis in the REST-KO and RC-control tumors. Hypoxia-inducible factor-1α (HIF-1α) was used to detect hypoxic areas. HIF-1α expression was significantly increased in both R1106 and R1606 REST-KO tumor tissues compared with RC-control tumors (Figure 6A,B).

A terminal deoxynucleotidyl transferase dUTP nick end labeling (TUNEL) assay showed that apoptosis was also significantly increased in the R1106 and R1606 tumors compared with control tumors (Figure 6C,D).

Taken together, these data indicate that inhibition of REST decreased the number of functional tumor vessels and increased vascular permeability, thus increasing tumor hypoxia and apoptosis.

## 3. Discussion

There is an urgent need for new therapeutic approaches for patients with ES, particularly those with metastatic or relapsed disease. Both the primary tumor and the metastases require a functional vascular system. Inhibiting vascular expansion compromises blood delivery, which hinders tumor growth and may therefore be an alternative therapeutic approach to treat tumors that do not respond or relapse after chemotherapy. Developing such therapies requires that we understand and define the molecular mechanisms that regulate tumor blood vessel expansion and functionality. This understanding is critical for identifying the potential targets that interfere with effective tumor vascular formation, blood flow, and oxygenation. We previously demonstrated that the REST gene is overexpressed in ES tumors and is regulated by EWS-FLI-1 [15,22]. In this study, we show that down-regulating REST using CRISPR/Cas9 impacts tumor growth and metastasis by targeting tumor blood vessel structure and function. While knockout of REST did not alter tumor cell proliferation in vitro, REST knockout reduced tumor growth and metastasis to the lung in vivo and altered tumor vascular morphology and function. Tumor vessels in the REST-knockout tumors had a punctate appearance with significantly decreased tumor vascular pericytes, decreased perfusion, and increased permeability. REST-knockout tumors also showed increased apoptosis and hypoxia. These results indicate that REST plays a critical role in ES vascular function, which in turn impacts the ability of ES tumors to grow and metastasize.

Our previous studies demonstrated that a portion of ES tumor vascular pericytes are derived from bone marrow progenitor cells [22], that VEGF165 was involved in the chemoattractant recruitment of these bone marrow cells to the tumor for formation of new blood vessels, and that the Notch-DLL4 signaling pathway regulated their differentiation into vascular pericytes [19,20,21]. Together with our previous report [15], the data presented here implicate REST in the regulation of the ES tumor vasculature.

We used CRISPR/Cas9 to knock out REST, creating two REST-KO clones with ≥80% reduction in REST expression, confirmed in vitro and in tumor samples (Figure 1A,B). Down-regulation of REST had no effect on cell proliferation in vitro but significantly inhibited tumor growth and metastasis in vivo and altered both the morphology and the function of the tumor vessels.

We also demonstrated a significant reduction in vascular pericytes and vascular perfusion in the REST-KO tumors. The ratio of pericytes to endothelial cells was significantly decreased. Pericytes play a critical role in blood vessel functionality and maturation, protection of endothelial cells, and regulation of endothelial cell viability and proliferation. Without pericytes, vessels are leaky and poorly perfused [23]. Indeed, loss of pericyte coverage in the REST-KO tumor vessels decreased vessel perfusion and increased vascular leakage. Pericyte depletion in tumor vessels also has previously been shown to induce tumor hypoxia [24]. Indeed, the decrease in tumor vascular pericyte coverage in the REST-KO tumors was associated with increased tumor hypoxia and apoptosis. Taken together, these data suggest that the vascular effects induced by REST KO, rather than an effect on cell proliferation, resulted in the reduced tumor growth and metastatic potential. These in vivo findings of decreased pericyte coverage of the tumor vessels in the REST-KO tumors are supported by our previous in vitro studies showing that REST regulated the expression of the pericyte markers desmin and NG2 in ES cells [15].

REST was originally described as a neuronal repressor gene that regulates neuronal stem cell differentiation [11,12]. Later data demonstrated that REST targets multiple genes in non-neuronal systems [13]. Recent studies indicated that REST modulates the vasculature in diffuse intrinsic pontine glioma (DIPG). REST has been shown to up-regulate the pro-angiogenic molecule gremlin-1 [25]. Inhibition of REST in DIPG caused a substantial decline in tumor vasculature, as measured by decreased CD31 and VEGFR2 staining, and reduced tube formation. Our results are in line with these findings but also show that the down-regulation of REST reduced both endothelial cells and the tumor vascular pericytes.

We have previously shown that tumors from ES patients express high levels of REST [15] and that REST controls the transdifferentiation of ES stem cells into pericytes in response to hypoxia [22]. Therefore, REST inhibition may compromise the ability of ES to contribute to the needed pool of vascular pericytes for tumor vascular expansion in an effort to recover from damage and cell death caused by chemotherapy or radiation therapy. A better understanding of the molecular mechanisms that are involved in tumor vascular formation and expansion, which impact both tumor growth and metastasis, may reveal new specific targets for anti-cancer therapy. Taken together with our previous work, our new results confirm that REST is a critical regulator of ES tumor vasculature integrity and function, pointing to REST as a new therapeutic target for ES.

## 4. Materials and Methods

### 4.1. Cell Lines

TC71 human ES cells were cultured in Dulbecco modified Eagle medium with 10% fetal bovine serum and authenticated by short terminal repeat fingerprinting at the Cytogenetics and Cell Authentication Core facility, The University of Texas MD Anderson Cancer Center. Human medulloblastoma cell line DAOY was purchased from the American Type Culture Collection. The cells were cultured in ATCC-formulated Eagle’s Minimum Essential Medium with 10% FBS according to the manufacturer’s instructions. All the cells were mycoplasma free as determined by the MycoAlert Mycoplasma Detection Kit (Lonza, Rockland, ME, USA).

### 4.2. Establishment of CRISPR/Cas9–REST-KO Cell Lines

Five pairs of DNA oligos were synthesized by Sigma-Aldrich (St. Louis, MO, USA) for REST-KO sgRNA expression, as follows:

REST-RY1F: caccgGTTATGGCCACCCAGGTAAT and REST-RY1R: aaacattacctgggtggccataacc; REST-RG2F: caccgAGACATATGCGTACTCATTC and REST-RG2R: aaacgaatgagtacgcatatgtctc; REST-RG4F: caccgCGCACCTCAGCTTATTATGC and REST-RG4R: aaacgcataataagctgaggtgcgc; REST-RG5F: caccgCAACAGTGAGCGAGTATCAC and REST-RG5R: aaacgtgatactcgctcactgttgc; REST-RG6F: caccgGTCTTCTGAGAACTTGAGTA and REST-RG6R: aaactactcaagttctcagaagacc.

Annealed double-stranded DNA fragments were cloned into a pX458-U6-chimeric_BB(+85)-CBh-NLS-hSpCsn1-2A-GFP vector (kindly provided by Dr. L. Copper from the Department of Pediatrics Research, MD Anderson Cancer Center, Houston, TX), and sgRNA expression clones were verified by restriction enzyme digestion and sequencing. TC71 cells were either transfected by a single sgRNA expression vector for single-nicking recombination or cotransfected by two sgRNA expression vectors for double-nicking recombination. After 48 h of transfection, single GFP expression cells were sorted into 96-well cell culture plates (one cell per well) with complete culture medium for continuing culture. Each single-cell culture was expanded, collected, and analyzed for REST expression by Western blotting analysis. REST-KO single-cell clones were selected for further in vitro and in vivo experiments. A random chosen clone with similar expression of REST as the parental TC71 cells was named TC71-RC-control and used as recombination control.

### 4.3. In Vivo Experiments

Four- to five-week-old athymic (T-cell deficient) nude mice were purchased from the National Cancer Institute and maintained in a specific pathogen-free animal facility approved by the Association for Assessment and Accreditation of Laboratory Animal Care International. The animal experiment protocol was approved by the Institutional Animal Care and Use Committee of MD Anderson Cancer Center (IACUC No. 00001400-RN01). TC71 parental cells, TC71-RC-control cells (with normal REST expression), R1106 REST-KO cells, and R1606 REST-KO cells in the mid-log growth phase were harvested by trypsinization. Cell suspensions (0.5 × 10^6^ cells in 0.01 mL of Hank’s balanced salt solution) were intra-tibially injected into the right leg of the nude mice. At day 24 after injection, the maximum diameters and 90° diameters of the tumors on the right leg were measured with a caliper and recorded. Tumor volume was calculated by the formula (*a*/2)^2^ × *b* minus the volume of the left leg, measured and calculated the same way, where a and *b* are the two largest diameters. The right leg was amputated at 4 weeks, and mice were maintained until 15 weeks after injection, when the mice were sacrificed, lungs were harvested, and visible lung tumor nodules were counted and recorded.

### 4.4. Blood Vessel Functional Assay Using DyLight 594–Labeled Tomato Lectin Perfusion

Lectin binds to glycoproteins located in the glycocalyx and in the basal membrane of endothelial cells. Intravenous injection of DyLight 594–labeled *Lycopersicon esculentum* (tomato) lectin is perfused through blood flow and bound inside the vessels. Functional vessels display red fluorescence color from bound DyLight 594–labeled tomato lectin, which can be visualized and quantified [26,27]. The endothelial marker CD31 shows total vessels, including both the perfused and non-perfused vessels. Thus, 24 days after the subcutaneous injection of RC-control or R1606 REST-KO cells (5 mice per group), each mouse was injected in the tail vein with 100 uL of DyLight 594–labeled tomato lectin (Vector Laboratories). The mice were euthanized 5 min later, and tumor tissues were separated and frozen in optimal cutting temperature medium (OTC) on dry ice. The tumor samples were sectioned and mounted in Vectashield HardSet with 4′,6-diamidino-2-phenylindole (DAPI) (VECTOR LABORATORIES, Burlingame, CA, USA). Perfused vessels were visualized as red tumor vessels under Leica fluorescence microscopy (Leica Microsystems, Buffalo Grove, IL, USA). The tumor tissues were then stained with FITC-labeled CD31 (green). Functional vessels were indicated by yellow fluorescence, which is the co-localization of FITC-labeled CD31 green and lectin-perfused red vessels. Co-localization of lectin and CD31 double-positive vessels and total CD31-positive vessels were quantified in 10 randomized fields from different tumor samples using SimplePCI software (Hamamatsu Photonics, Bridgewater, NJ, USA). The percentage of perfused vessels among total vessels was calculated.

### 4.5. Dextran-FITC Vessel Permeability Assay

Dextran labeled with FITC (Sigma-Aldrich) was used to assess vessel permeability. Normal blood vessels are not permeable to substances with high molecular weight, including dextran (2 × 10^6^ kDa); when vessel permeability increases, dextran leaks to the tissue outside of the vessels [28]. Therefore, 100 μL of FITC-labeled dextran (10 mg/mL) was intravenously injected into the mice bearing RC-control, R1106 REST-KO, or R1606 REST-KO tumors 5 min before the mice were sacrificed (three groups tumor-bearing mice, 5 mice per group). The harvested tumor tissues were then frozen and sectioned, and CD31 immunofluorescence staining was performed. Texas Red-conjugated anti-rat IgG was used as the secondary antibody. The FITC-labeled dextran area (green) outside of vessels (red, CD31-positive) indicated leaky blood vessels. The FITC-positive area was quantified in 10 different microscopic fields from different tumor samples using SimplePCI software (Hamamatsu Photonics, Bridgewater, NJ, USA), and average positive area was calculated.

### 4.6. Western Blotting

Different CRISPR/Cas9–REST-KO clones, DAOY human medulloblastoma cells, and CRISPR/Cas9 RC-control cells (as positive control) were cultured in 100-mm diameter dishes. Cell lysate was collected. The protein (40 μg each lane) was loaded onto 8% sodium dodecyl sulfate–polyacrylamide gel. Specific protein bands were detected with anti-human REST (Millipore, Burlington, MA, USA) and β-actin (Sigma-Aldrich, St. Louis, MO, USA) antibodies. Densitometry analysis was performed, and the values were normalized with β-actin as loading control.

### 4.7. Immunofluorescence Staining

Frozen tumor sections were fixed with acetone and chloroform. The sections were incubated with REST antibody (Millipore); NG2, HIF-1α, or Ki67 antibody (Abcam, Cambridge, UK); or rat anti-mouse CD31 antibody (BD Biosciences, San Jose, CA, USA). Anti-rabbit cyanine 5, anti-rat Texas Red, or FITC was used as the secondary antibody. All stained slides were analyzed by fluorescence microscopy (Leica Microsystems). Relative expression was quantified in at least five different microscopy fields from different tumor samples using SimplePCI software (Hamamatsu), and the average expression was quantified.

### 4.8. Statistical Analysis

All values are reported as means ± SD (standard deviation). A two-tailed Student *t*-test was used to statistically evaluate the in vitro experimental results. *p* < 0.05 was considered statistically significant. One-way ANOVA analysis of variance was used for statistical analysis of animal experiment results by Graphpad Prism 8 software (San Diego, CA, USA).

## 5. Conclusions

In conclusion, the inhibition of REST reduced tumor vessel pericytes and altered vasculature morphology and functionality. Tumor vessels from REST-KO tumors were punctate and poorly perfused and showed increased permeability and leakiness. The vascular changes were associated with increased tumor hypoxia and apoptosis, as well as decreased tumor growth and metastasis. These results indicate that REST may be a key regulator of ES vascular development and expansion and that inhibiting REST could compromise the ability of the tumor to grow and metastasize by compromising the tumor’s vascular function. Inhibiting vascular function by blocking REST may also compromise the tumor’s ability to recover and regrow following chemotherapy or radiation therapy, as the oxygen and nutrients required for recovery may not be delivered. Therefore, the critical vascular functions that are controlled by REST provide a rationale for considering REST as a potential target for the treatment of ES.

## Figures and Tables

**Figure 1 cancers-12-01405-f001:**
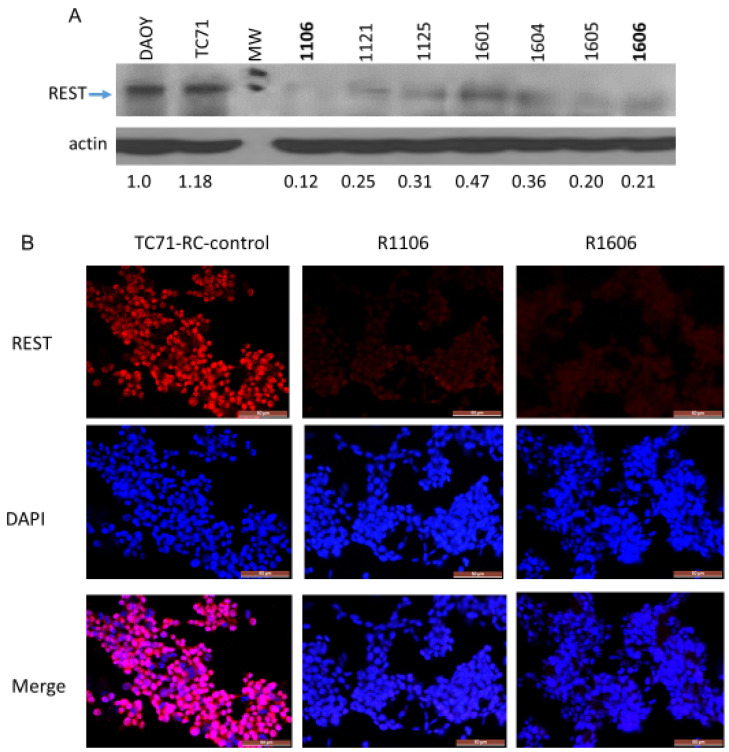
Assessment of REST in the TC71 CRISPR/Cas9–REST-KO clones. (**A**) Western blot analysis of REST protein expression in several CRISPR/Cas9–REST-KO clones. Densitometry indicated REST protein expression was reduced by at least 80% in the R1106 and R1606 clones compared with the TC71 parental cells. DAOY human medulloblastoma cells were used as the positive control. (**B**) Immunofluorescence staining of REST expression in RC-control cells and the R1106 and R1606 clones. REST expression (red) was significantly down-regulated in R1106 and R1606 cells compared with the RC-control cells. DAPI was used for nuclear staining. Scale bar: 50 µm.

**Figure 2 cancers-12-01405-f002:**
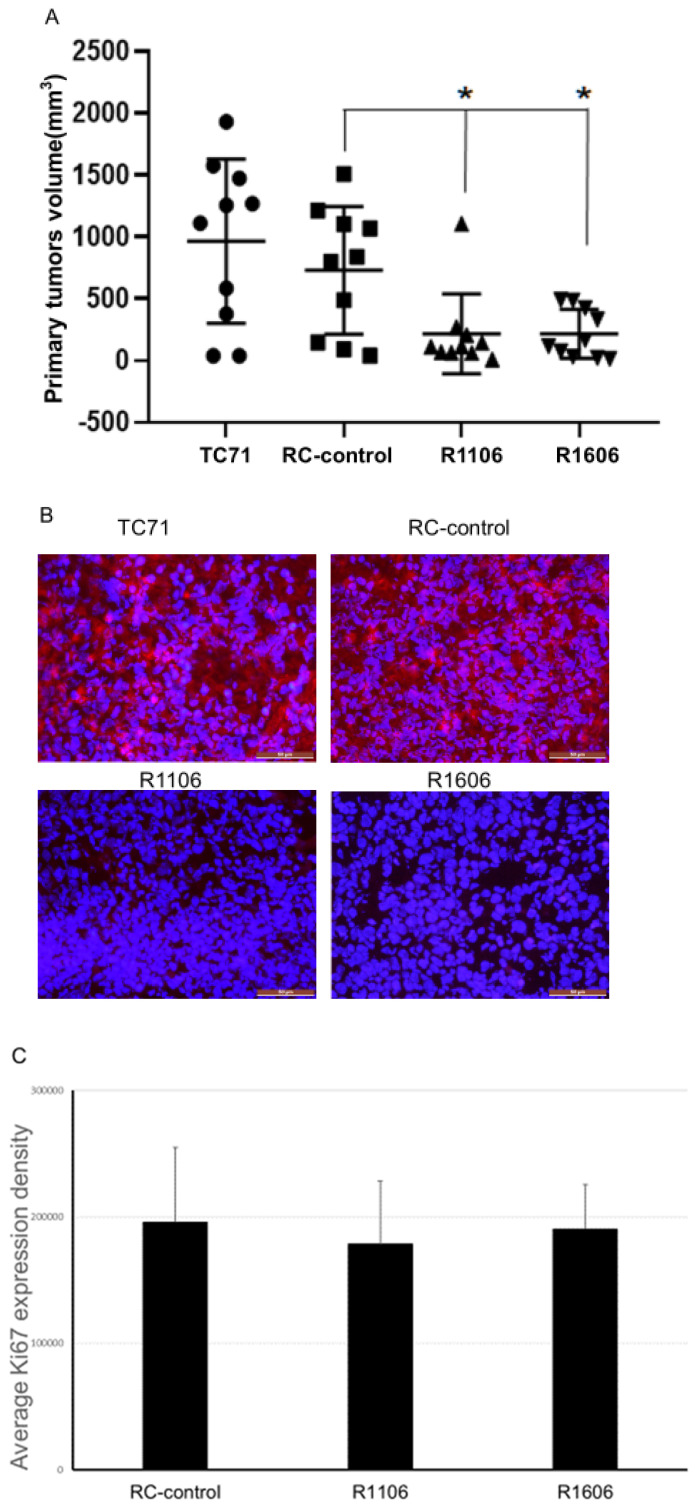
Down-regulation of REST inhibited tumor growth. (**A**) Totals of 5 × 10^5^ TC71 parental, RC-control, R1106, or R1606 cells were injected into the tibias of nude mice (10 mice per group). Tumor size was measured 4 weeks later, and the average tumor volume was calculated. One-way analysis of variance (ANOVA) showed a significant difference in tumor volume among groups (*p* = 0.001) or between groups (* *p* < 0.05). (**B**) Immunofluorescence staining for REST expression in the different tumor tissues confirmed that REST was down-regulated in the R1106 and R1606 tumor samples. Scale bar: 50 µm. (**C**) The Ki67 cell proliferation marker was used to assess cell proliferation. Ki67 expression was quantified in the different tumor samples by PCI software. Ki67 expression was not significantly different in the R1106 and R1606 tumor tissues compared with the RC-control tumors. (**D**) Leg amputations were performed at 4 weeks after tumor cell intra-tibial injection, and mice were maintained until 15 weeks after injection, when mice were killed, lungs were harvested, and visible lung tumor nodules were recorded. One-way ANOVA analysis of variance showed a significant difference in lung metastasis among groups (*p* = 0.0197) or between groups (* *p* < 0.05).

**Figure 3 cancers-12-01405-f003:**
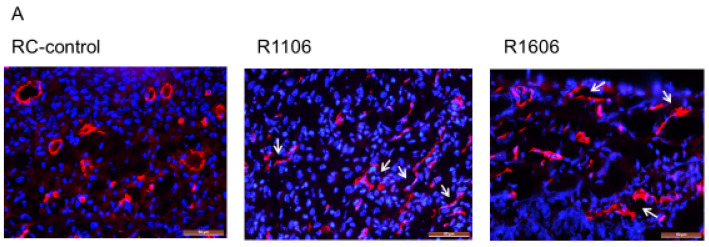
Inhibition of REST decreased the number of vascular endothelial cells and pericyte coverage in tumor vessels. (**A**) The endothelial marker CD31 (red) was used to identify tumor vascular structure by immunofluorescence staining. White arrows indicate the punctate lumens. Scale bar: 50 µm. (**B**) Double staining for the pericyte marker NG2 (green) and endothelial marker CD31 (red) was performed in the different tumor tissues to assess vascular pericytes coverage. NG2 expression and NG2 + CD31 dual-positive vessels were decreased in the R1106 and R1606 tumors compared with the RC-control tumors. White arrows indicate double positive staining (yellow color). Scale bar: 50 µm. (**C**) Mean NG2 expression in each of the tumor groups was quantified. Bars represent standard deviation. * *p* < 0.01. (**D**) The ratio of NG2 to CD31 was calculated in each tumor. Bars represent standard deviation. * *p* < 0.01.

**Figure 4 cancers-12-01405-f004:**
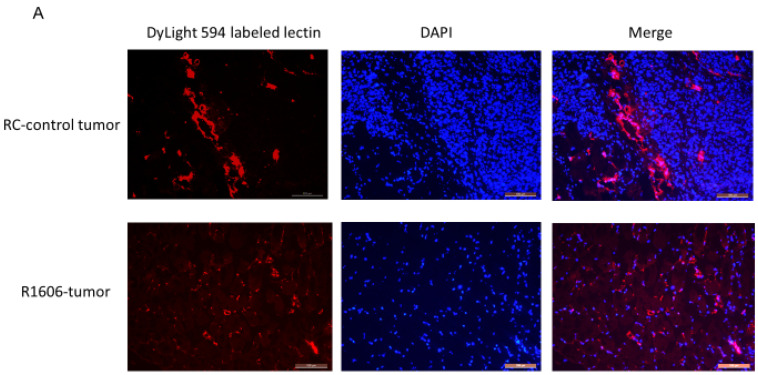
Down-regulation of REST reduced tumor vascular perfusion. (**A**) The mice (two groups tumor-bearing mice, 5 mice per group) were injected with DyLight 594–labeled *Lycopersicon esculentum* (tomato) lectin, then euthanized after 5 min. Tumor tissue sections were analyzed for tumor vessel perfusion (red). Scale bar: 100 µm. (**B**) The average number of perfused tumor vessels quantified for each group. Bars represent standard deviation. * *p* < 0.01. (**C**) CD31 immunofluorescence staining was performed in the lectin-perfused samples. Co-localization of CD31 (green) and lectin (red) vessels indicated the perfused vessels (yellow) as indicated by white arrows. Scale bar: 100 µm. (**D**) Functional vessels (yellow) and total perfused vessels (red) were quantified. The percentage of functional vessels was calculated in each group. Bars represent standard deviation. * *p* < 0.01.

**Figure 5 cancers-12-01405-f005:**
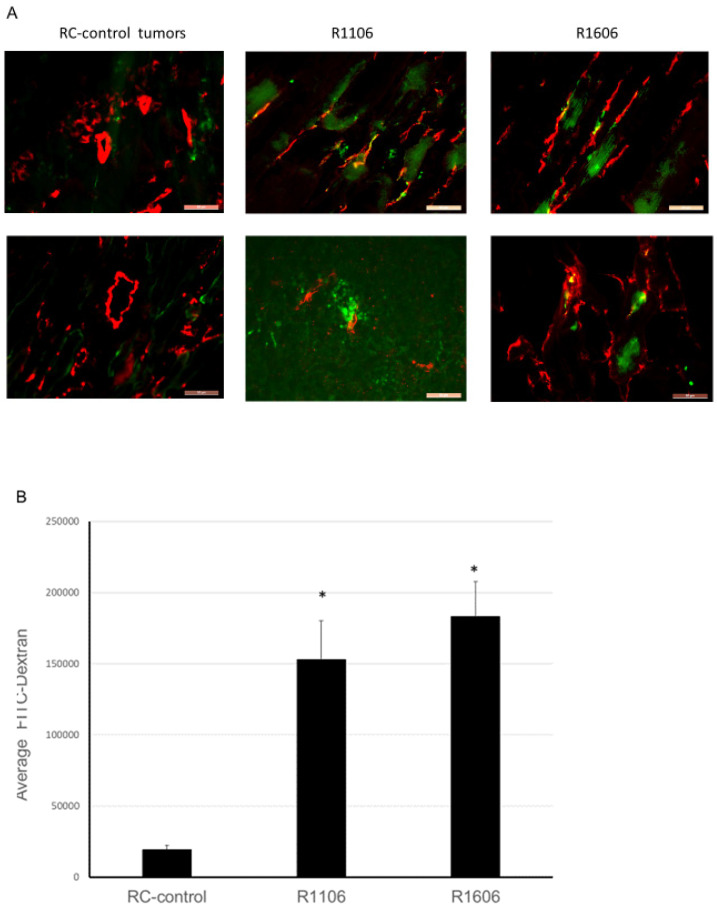
Down-regulation of REST increased vessel permeability. (**A**) Dextran-FITC vessel permeability assay and CD31 staining were performed in the different tumor samples from tumor-bearing mice (three groups of mice, 5 mice per group). CD31 expression indicates tumor vessels, and FITC-labeled dextran (green) indicates vascular leakage due to increased permeability. Scale bar: 50 µm. (**B**) The averages of FITC-labeled dextran in the different tumor samples were quantified. Bars represent standard deviation. * *p* < 0.01.

**Figure 6 cancers-12-01405-f006:**
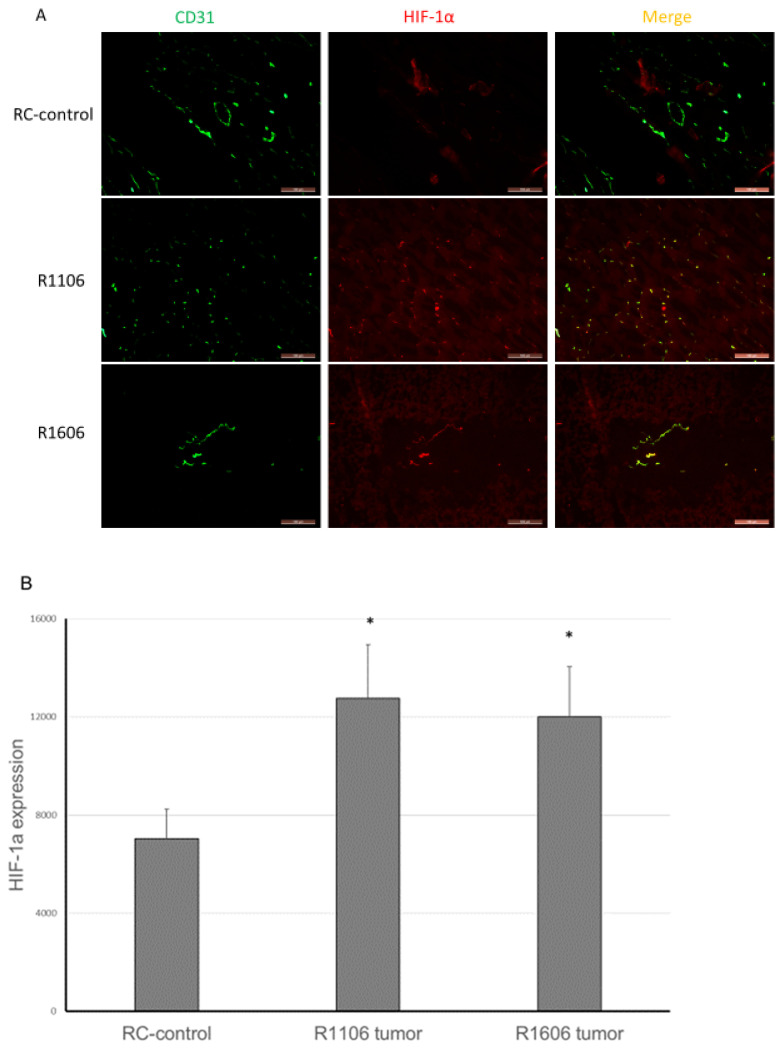
REST-KO increased tumor hypoxia and apoptosis. (**A**) Hypoxia marker HIF-1α expression was detected by immunofluorescence staining (scale bar: 100 µm). (**B**) The average HIF-1α expression was quantified by assessment of five random fields in each group. Bars represent standard deviation. * *p* < 0.05. (**C**) TUNEL assay was performed to detect apoptosis (scale bar: 50 µm), and (**D**) the average apoptosis area was quantified by assessment of five random fields per group. Apoptotic cells were significantly increased in R1106 and R1606 REST-KO tumors compared with RC-control tumors. Bars represent standard deviation. * *p* < 0.05.

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
