# Peer review of "Neuronal Repressor REST Controls Ewing Sarcoma Growth and Metastasis by Affecting Vascular Pericyte Coverage and Vessel Perfusion"

_cancers, 2020, doi:10.3390/cancers12061405_

Round 1

Reviewer 1 Report

The authors clearly demonstrate how the inhibition of REST in ES tumor  to decrease tumor growth in vivo.
However, even if the work is well written, the work  requires  more experiments.
The images of apoptosis and HIF expression in the tumor tissue has to be shown and the concept of hypoxia has to be clarified. It is well known that an activation of HIF-1 alpha leads to cell survival which is regulated by a lot of genes that affect cell metabolism, angiogienesis and metastases.HIF-1 and Mechanisms of Hypoxia Sensing PMID: 11248550

How  do explain the author this concept. It would be advisable to continue treating the mice longer by inhibiting REST for longer periods to see if the increase in HIF -1 leads to  relapse of disease, which is also very common in Ewing Sacoma. Furthermore, in order to understand these concepts, the VEGF level should be shown when HIF-1 rises. I would also require a colocalization with immunofluorescence staining of HIF1 and CD31 in the endothelial cells surrounding the tumor. In the discussion the author should stress the concept of Hypoxia and HIF-1 activation.

Minor points

The Figures has poor resolutions

Author Response

Response to Reviewer #1 Comments:

We appreciate all the opinions, comments and suggestions from the reviewer. Detailed replies are as follows:

Reviewer #1 Specific comment #1:

Reviewer requested that “the images of apoptosis and HIF expression in the tumor tissue has to be shown”.

We added the HIF1α staining images together with CD31 staining as new figure 6A. The original figure 6A for quantification comparison of HIF1α expression is subsequently renamed as figure 6B.

We also added the TUNEL assay staining images for apoptosis together with DAPI staining as new figure 6C. The original figure 6B for quantification comparison of TUNEL assay staining is subsequently renamed as figure 6D.

All related number of figures in the text are modified accordingly.

Reviewer #1 Specific comment #2:

Reviewer asked to clarify the concept of hypoxia.

Our data indicated that inhibition of REST altered tumor vessel morphology (decreased pericyte coverage) resulting in leaky or broken vessels. Therefore, blood flow and oxygen cannot be properly delivered to the tumor area, which subsequently induced tumor hypoxia. Thus, the tumor hypoxia resulted from the vascular insufficiency which led to tumor cell apoptosis. Upregulated HIF1α was the result of hypoxia caused by the decreased pericyte coverage and leaky or broken vessels. We interpret this data to support the concept that HIF1α upregulation is a late effect.

Reviewer #1 Specific comment #3:

Reviewer suggests continuing treating the mice longer by inhibiting REST for longer periods to see if the increase in HIF -1 leads to relapse of disease, which is also very common in Ewing Sarcoma.

In this study, we are focused on both primary tumor and lung metastasis. For this purpose, at week 4 after the intra-tibia injection, we amputated the primary tumors completely by removing the tumor bearing left leg from the hip joint and assessed the mice for lung metastasis 15 weeks later. During these 15 weeks, we did not observe relapse of the primary tumor in the surrounding tissues.

Reviewer #1 Specific comment #4:

Reviewer suggests we show the VEGF level when HIF-1 rises.

In our previous report (Reference 15), we checked the VEGF level after we downregulated REST by siRNA in TC71 cells. We observed that after REST down regulation, HIF1a expression is upregulated, but there is no significant different in VEGF expression between TC71 and siRNA transfected tumor cells. Therefore, we didn’t analyze the VEGF level in this report.

Reviewer #1 Specific comment #5:

Reviewer requested double immunofluorescence staining of HIF1 and CD31 in the endothelial cells surrounding the tumor.

Please see response to comment #1 above and new Fig. 6A for double immunofluorescence staining of HIF1 and CD31.

Reviewer #1 Specific comment #6--Minor points:

The Figures has poor resolutions. 

We have uploaded the high-resolution figures as PPT file to the “Cancers” website together with this revision.

Reviewer 2 Report

  In this manuscript the authors have shown very interesting results of their research on the impact of neuronal repressor REST on Ewing sarcoma tumors. This is a continuation and extension of their previous studies. Present experiment, with CRISP/Cas9 technology using to knock out REST expression in ES cells, have been designed and carried out correctly. The final conclusions are fully supported by the results. They clearly show that REST knockout diminishes the ES tumors ability  to grow and metastasize due to an effect on the tumor vasculature rather than a direct effect of the tumor cells. This is consistent with authors’ results obtained before using different methods.

As authors have stated, the treatment results of ES, particularly its disseminated form, are not satisfactory and they haven’t been improved for almost three decades. Thus, the results of this study are very promising and give some hope that REST can be a potential, molecular target for an ES therapy. So, I fully recommend this manuscript for publication. I have no major issues and only few minor ones: - the last part of introduction section – there are just conclusions and, in my opinion, it is unnecessary to provide them twice.
- the number of mice tested in the experiments in vivo was provided only in the description of the Figure 2 and in the point 4.4- “Blood vessel functional assay…”. It  should also be added to the other parts of the experiment in the Material and Methods section.
- point 2.4 – Down-regulation of REST reduces tumor vascular perfusion…. Please explain why the line R1106 was excluded from this part of your experiment? 

I also think that an application of e.g spectral and power Doppler ultrasounds used to measure blood flow velocities and ultrasound contrast agents for angiogenesis assessment in tumors in vivo could have been an interesting addition to this experiment.

Author Response

Response to Reviewer #2 Comments:

We appreciate all the opinions, comments and suggestions from the reviewer, detailed replies as follow:

  1. Reviewer suggested the last part of introduction section is just conclusions and is unnecessary to provide twice.

    As reviewer suggested, we remove the redundant sentences of the last part of introduction section.
  2. Reviewer suggested the number of mice tested in the experiments in vivo should also be added to the other parts of the experiment in the Material and Methods section.

    We put the number of mice used in all the experiments in vivo in all the related description parts of the experiment including in the description of the Figure 2 and in the point 4.4- as well as in the Material and Methods section, as suggested by the reviewer.
  3. Reviewer asked explanation why the line R1106 was excluded from experiment 2.4 – Down-regulation of REST reduces tumor vascular perfusion:

    Both REST-KO line R1106 and R1606 gave similar results in the primary tumor growth, lung metastasis potential and vascular morphology, thus out of both REST-KO lines, we selected to use R1606 for the vascular perfusion study.
  4. Reviewer suggested to use power Doppler ultrasounds to measure blood flow velocities or using ultrasound contrast agents for angiogenesis assessment in tumors in vivo are could have been an interesting addition to this experiment.

This is an excellent suggestion, but detailed measurements of blood flow velocities and tumor angiogenesis is beyond the scope of the present manuscript as we are focusing on the effect of REST knockout on tumor growth, metastasis, and vascular morphology.

Round 2

Reviewer 1 Report

In my opinion,
the work can be accepted with new revisions.